# Effects of different blood flow rates on filter and circuit life in non-anticoagulation CRRT: Protocol for a three-arm, single-blind randomized controlled trial (the Flow-CRRT Study)

**Caihong Liu[1], Qiongxing Bu[1], Fang Wang[2], Wei Wei[1], Yongxiu Huang[1], Jinglei Ren[1], Jay L. Koyner[3], Ling Zhang[1], Yuliang Zhao[1]\***

**1** Department of Nephrology, Institute of Kidney Diseases, West China Hospital, Sichuan University, Chengdu, China, **2** Department of Nephrology, West China Hospital of Sichuan University/West China School of Nursing, Sichuan University, Chengdu, China, **3** Department of Medicine, University of Chicago, Chicago, United States of America

\* zhaoyuliang@scu.edu.cn

## Abstract

### Background

Non-anticoagulation is a commonly used strategy in continuous renal replacement therapy (CRRT) among patients with high-bleeding risk. However, the optimal blood flow rate (BFR) to maximize filter and circuit life remains uncertain. This study is designed to elucidate the impact of different BFRs on the durability of filters and circuits in CRRT without anticoagulation.

### Methods

This single-center, prospective, three-arm, single-blind, randomized controlled trial (RCT) will involve adult patients requiring non-anticoagulation continuous venovenous hemodiafiltration (CVVHDF). A total of 486 filters and circuits will be enrolled and randomly assigned to one of three BFR groups: low (150 mL/min), medium (200 mL/min), or high (250 mL/min) BFR group. The outcomes will be analyzed by both intention-to-treat analysis and per-protocol analysis. The primary outcome is filter and circuit life, which is defined as the time from CRRT initiation to CRRT termination due to extracorporeal circuit clotting or other reasons, alongside the proportion of patent circuits at 24, 48, and 72 hours. Secondary outcomes encompass clinical outcomes and potential adverse events such as bleeding and hemodynamic alterations.

### Discussion

This study is aiming at comparing the filter and circuit life under different BFR levels during CVVHDF without anti-coagulation. The results may add knowledge to the

**Data availability statement:** No datasets were generated or analysed during the current study. All relevant data from this study will be made available upon study completion.

**Funding:** The author(s) received no specific funding for this work.

**Competing interests:** The authors declare that they have no competing interests.

optimal BFR to prevent extracorporeal circulation clotting and prolong filter and circuit life in non-anticoagulation CRRT.

## Trial registration

The study has been registered at https://www.chictr.org.cn (ChiCTR2400087819).

## 1. Introduction

Continuous renal replacement therapy (CRRT) is a pivotal blood purification technique, renowned for its role in eliminating excess water, toxins, inflammatory factors, and autoantibodies from the circulation. It has become increasingly used in the management of a spectrum of critical conditions such as acute kidney injury (AKI) and sepsis [1]. The successful performance of CRRT is contingent upon the unobstructed extracorporeal circulation, thus maintaining it for at least 24 hours. Clotting of the filter and circuit can prematurely terminate therapy, diminishing treatment efficacy, and increasing the risk of blood loss, healthcare workload, and economic burden [2–4]. For decades, studies have focused on optimized anticoagulation strategies to extend the life of extracorporeal devices, such as heparin and regional citrate anticoagulation (RCA) [5,6]. Nevertheless, coagulation-free CRRT is inevitable, especially in patients with high bleeding and those with contraindications of available anti-coagulants [7]. A large multinational study reported that non-anticoagulation CRRT accounted for 24% of all CRRT treatments [8]. However, it is reported that filter and circuit life was significantly shorter for the non-anticoagulation strategy in comparison with RCA-CRRT (14.3 vs. 44.9 hours, $P < 0.001$), though no significant disparity in CRRT efficacy has been observed [9].

Beyond anticoagulants, non-pharmacological interventions to prolong filter and circuit life, including the choice of therapy modalities, blood flow rates (BFR), catheter sites and types have been investigated [10]. Evidence showed that continuous veno-venous hemodialysis (CVVHD) and continuous veno-venous hemodiafiltration (CVVHDF) might prolong filter lifespan compared with continuous veno-venous hemofiltration (CVVH) [11,12], while no difference was observed between CVVHD and CVVHDF groups [13]. BFR is a key, easily adjustable parameter in CRRT, and clinical practice varies widely, with BFRs ranging from 80 to over 300 mL/min globally [14,15]. The Kidney Disease: Improving Global Outcomes (KDIGO) guidelines suggest that BFRs between 150–250 mL/min are typically prescribed for CRRT modalities such as CVVH and CVVHDF, but offer no evidence-based recommendations for optimal values [16]. While BFRs > 200 mL/min have been empirically recommended to maintain circuit patency [17], more recent studies have suggested that very high BFRs (e.g., > 250 mL/min) may actually shorten circuit life, possibly due to increased circuit pressure [13,18]. A randomized controlled trial (RCT) comparing BFRs of 150 and 250 mL/min in CRRT found no significant difference in circuit clotting risk, irrespective of anticoagulation strategy [19]. Given the relatively low quality, substantial heterogeneity, and controversial findings among previous studies, there is insufficient evidence to draw a solid

conclusion on optimal BFR in non-anticoagulation CRRT. Moreover, while clinicians often choose among standard BFR settings (150, 200, or 250 mL/min), no study to date has systematically compared circuit lifespan across these three levels in non-anticoagulated CRRT. In this parallel-group, single-blind, three-arm RCT, we aimed to compare the impacts of different BFRs (150 mL/min vs. 200 mL/min vs. 250 mL/min) on circuit life and other secondary outcomes in patients receiving non-anticoagulation CRRT. This study aims to inform practice where no evidence currently guides BFR selection.

## 2. Methods and analysis

### 2.1. Ethics and dissemination

The study is conducted in compliance with the Declaration of Helsinki and has received ethical approval from the Institutional Review Board of West China Hospital (Approval No: 2024—918). It has been registered at https://www.chictr.org.cn (ChiCTR2400087819) on 5 August 2024. It is important to note that neither participants nor the public will be involved in the study's design, conduct, or dissemination. We will publish the results of our research in medical journals after the study. The study data will be available upon request, ensuring transparency and accessibility. The enrollment of patients is scheduled to commence at a university-affiliated medical center in September 2024. This study is ongoing now. The participant recruitment will be completed in March 2025. Throughout the trial, trained clinical staff will closely monitor adverse events, including hypotension, arrhythmias, bleeding, and other potential complications. If one group exhibits a significantly higher incidence of serious adverse events related to treatment (non-anticoagulation CRRT or BFR setting), such as hypotension, malignant arrhythmia, or death, compared to the other two groups, or if the incidence exceeds predefined safety thresholds, termination of the intervention for that group will be considered immediately. Additionally, an independent Data Safety Monitoring Committee (DSMC) will regularly review trial data to ensure patient safety.

### 2.2. Study design

This will be a monocenter, prospective, parallel-group, single-blind, three-arm RCT designed to evaluate the impact of BFRs on filter and circuit life during anticoagulation-free CRRT. After obtaining informed consent, 486 sets of filters and circuits will be consecutively recruited and randomized to low BFR (150 mL/min) group, medium BFR (200 mL/min) group, or high BFR (250 mL/min) group. Given that a single patient may contribute multiple filters and circuits, each filter or circuit is treated as a distinct unit for randomization within the study. The allocation concealment will be maintained such that the participants are blinded to the treatment assignment: (1) Allocation Concealment: The CRRT machines and settings will be managed exclusively by the trained CRRT staff, who will not disclose BFR assignments to patients or their families. (2) Visual Blinding: The BFR setting on the CRRT machine screen will be obscured or made non-interpretable to patients to prevent inadvertent unblinding. The clinical teams are not masked as the BFR is shown on the machine. Adhering to the Standard Protocol Items: Recommendations for Interventional Trials (SPIRIT) 2013 Statement, we have outlined the trial design and protocol in detail (Fig 1).

### 2.3. Study objectives

The principal objective of this RCT is to ascertain the impact of varied BFRs on the longevity of filters and circuits in the context of anticoagulation-free CRRT, including the filter and circuit life and the proportion of patent circuits living up to 24 hours, 48 hours, and 72 hours. The secondary objectives encompass the comparisons regarding the blood pressure changes, extracorporeal circulation pressure, patients' in-hospital mortality, hospital and ICU duration, invasive mechanical ventilation and dialysis-dependency rate, and the incidence of adverse events like bleeding.

### 2.4. Study participants

Adult patients who require CRRT at a university-affiliated medical center are screened as potential participants in this RCT. The inclusion criteria are as follows: a) aged over 18 years; b) scheduled to receive non-coagulation CRRT; and c)

Schedule of enrollment, interventions, and assessments according to the Standard Protocol Items: Recommendations for Interventional Trials (SPIRIT) guideline.

| | Enrollment | Allocation | Postallocation (CRRT) | | | | | | closeout |
|---|---|---|---|---|---|---|---|---|---|
| **Timepoints** | | | before 30 mins | initiation | 24-hour | 48-hour | 72-hour | termination | |
| **Enrollment** | | | | | | | | | |
| Eligibility screen | X | | | | | | | | |
| Informed consent | X | | | | | | | | |
| **Allocation** | | X | | | | | | | |
| **Interventions** | | | | | | | | | |
| BFR | | | X | X | X | X | X | | |
| Post-dilution | | | X | X | X | X | X | | |
| **Assessment** | | | | | | | | | |
| Clotting | X | | | | | | | | |
| Bleeding | | | X | X | X | X | X | X | |
| Laboratory tests | | | X | X | X | X | X | X | |
| **Outcome** | | | | | | | | | |
| Filter and circuit life | | | | | | | | X | |
| Length of hospitalization | | | | | | | | X | |
| CRRT-related outcomes | | | | | | | | X | |
| Mortality | | | | | | | | | X |
| Kidney-related outcomes | | | | | | | | | X |

Abbreviations: CRRT, continuous renal replacement therapy; BFR, blood flow rate.

**Fig 1. Schedule of enrollment, interventions, and assessments according to the Standard Protocol Items: Recommendations for Interventional Trials (SPIRIT) guideline.**

provision of informed consent by the patient or their legally authorized representative. Patients are excluded for the following reasons: a) received systemic anti-coagulation for indications other than CRRT (for example ECMO); b) unable to perform CRRT according to the pre-set BFR due to hemodynamic instability, vascular access dysfunction (CRRT machine alarm and blood pump shutdown occurs due to poor catheter function) or other reasons; c) pregnant women. Nephrologists specialized in CRRT will be responsible for screening when patients are considered requiring CRRT without coagulation by a joint team of nephrologist and ICU physician. All the eligible patients or his/her proxy will be thoroughly informed clearly of the objective, procedure, expected benefits, and possible risks of participating in this study. Both the patient and the treating physician are required to sign the informed consent form, signifying their mutual understanding and agreement. Confidentiality of all patient information will be maintained, and patients have the right to withdraw from the study at any time without incurring any penalties or prejudice.

## 2.5. Randomization

The random number sequences are generated with Python by an independent statistician, ensuring the utmost impartiality and integrity. Upon confirmation of eligibility, participants will be randomly allocated in a 1:1:1 ratio to one of the three study groups: low, middle, or high BFR. The respective BFRs for these groups during non-anticoagulation CRRT will be set at 150 mL/min, 200 mL/min, and 250 mL/min. The allocation results will be monitored by an independent coordinator and blinded to the patients.

## 2.6. Intervention

All the patients involved in this study will receive non-anticoagulation CVVHDF in post-dilution mode. The CRRT machine and catheter type, therapeutic dosage, filtration rate, and vascular access are not pre-specified and may vary according to clinical standards and patient-specific requirements. The BFR for extracorporeal circulation will be standardized into three levels across participants. It is stipulated that the BFR may be adjusted downward in a stepwise manner only in instances where the circuit experiences excessive pressure or triggers pressure alarms due to vascular access insufficiency.

However, if the team decides that a lower BFR is not acceptable, the catheter will be adjusted or reimplanted to guarantee the predefined target. In instances where a patient discontinues extracorporeal circulation due to unplanned disruptions like examination/therapy trips, transference to other departments/hospitals during CRRT, the duration of filter/circuit utilization shall be meticulously documented.

## 2.7. Outcome measures

The primary outcome is filter and circuit life, defined as the time interval from the initiation to the termination of a single CRRT session for any reasons as well as for non-electively ceased filter and circuit due to clotting. We will also compare the proportions of extracorporeal circulation achieving 24-hour, 48-hour, and 72-hour lifespans, with specific attention to the filters and circuits suspension owing to coagulation.

The secondary outcomes include 1) patients' clinical outcomes: in-hospital mortality, length of hospital/ICU stay, and invasive mechanical ventilation and dialysis-dependency rate; 2) hemodynamic parameters such as blood pressure changes during CRRT; 3) adverse events like bleeding.

## 2.8. Sample size calculations

According to our retrospective analysis enrolling CRRT patients with non-coagulants in West China Hospital of Sichuan University, the median filter and circuit life of no-anticoagulation CRRT within our unit was $28.09 \pm 21.15$ hours for a BFR of 200 mL/min, whereas $34.17 \pm 16.36$ hours for a BFR of 250 mL/min. By employing Power analysis and sample size software (version 11.0, PASS, NCSS LCC, Kaysville, UT, USA), we calculated the sample size with a two-sided type I error of 0.05 and 90% power. One hundred and fifty-three extracorporeal circuits, will be needed in each group. Accounting for a dropout rate of 5% where the patient/proxy withdraw inform consent or the physician considers the patient unsuitable to continue the study, a total of 486 filters or circuits are to be included in this RCT.

## 2.9. Statistical analysis

Personal information (e.g., name, telephone number) will be kept confidential. Statistics will be presented as the means ± standard deviations (SDs) or medians with interquartile ranges (IQRs) for continuous variables and percentages for categorical variables. For missing data, the multiple imputation method will be used as appropriate, which allows for a more robust and statistically valid approach by incorporating uncertainty. Differences between the two groups will be measured by t test for continuous variables with a normal distribution, the Mann–Whitney U test for variables with a nonnormal distribution, and the $\chi 2$ test or Fisher's exact test for categorical variables.

Time to filter or circuit coagulation will be compared between groups by the use of Kaplan-Meier curves and log-rank tests. Univariable analysis will be conducted to evaluate the impact of potential confounding factors (such as age, body mass index, laboratory parameters, and filtration fraction) on the filter and circuit survival. Then multivariable Cox regression model will be adjusted to estimate the independent risk factors. The results will be expressed as hazard ratio (HR) and 95% confidential interval (95% CI). We will use the Schoenfeld residual test to verify the assumption of proportional hazards in the Cox analysis. In cases of non-proportionality, stratified Cox models or incorporating time-varying effects will be considered as appropriate. In addition, logistic regression will be conducted to identify the potential risk factors for the incidence of pre-mature extracorporeal circulation coagulation, which will be presented by odds ratio (OR) and 95% CI. Subgroup analyses will be conducted to separately evaluate the effects of BFR on filter lifespan and circuit lifespan, as well as based on the type of filter used. Of note, the sample size might limit the ability to control for numerous covariates, even with randomization. Therefore, we will prioritize the independently protective or risk factors identified by a retrospective cohort study we previously conducted, including filter type, substitute fluid rate, patients' coagulation function, hematocrit level, and whether exposure to non-CRRT anti-coagulation agents or concomitant with hyperlipidemia to control for. Although if it is balanced as expected, we will conduct a sensitivity analysis to examine the robustness of our primary findings.

 

Given that patients may receive multiple CRRT sessions, resulting in clustered data, we will use a mixed-effects model to account for within-patient correlation, treating individual filters as the unit of analysis and patients as random effects. A random intercept model will capture different baseline risks of filter failure across patients. The primary fixed effect will be BFRs, with patient-level covariates such as age, baseline coagulation function, and renal function included to assess their independent associations with filter and circuit life. However, current data may not provide sufficient power to reliably estimate all variance components in the proposed complex random-effects structure. To address this methodological challenge while maintaining scientific rigor, we will implement a stepwise modeling approach: Initially, we will attempt to fit the pre-specified maximal model incorporating all planned random effects. Should we encounter convergence issues or observe unstable variance estimates (operationally defined as standard errors exceeding the mean estimate), we will systematically simplify the model by first removing random slopes for less critical covariates while preserving random intercepts for key clustering variables like study sites. Throughout this process, we will conduct comprehensive sensitivity analyses to evaluate the impact of model simplifications on both model fit statistics (AIC/BIC) and the stability of fixed-effects estimates, with particular attention to our primary exposure variables. All model adaptations, along with their justifications and validation results, will be thoroughly documented in the supplementary materials to ensure full transparency. Poisson regression will be used to model the count data for coagulation frequency, assuming that the mean and variance are equal. If the variance substantially exceeds the mean, a phenomenon known as overdispersion, negative binomial regression may be considered, since it allows for greater flexibility by incorporating a dispersion parameter. In both models, patient ID will be a random effect to account for repeated measures, and rate ratio (RR) with 95% CI will be estimated for the impact of covariates on filter and circuit survival.

Both intention-to-treat (ITT) analysis and per-protocol (PP) analysis will be performed. For ITT analysis, once enrolled in the study, all data will be analyzed according to the group randomly assigned, regardless of possible protocol violations (e.g., changes in anticoagulation modality or BFR). For PP analysis, only CRRT sessions that finished the treatment as planned will be included in the analyses. All the statistical analyses were performed via R software (version 4.3; R Foundation for Statistical Computing, Vienna, Austria). A *P* value of < 0.05 will be considered to indicate statistical significance.

## Discussion

The FLOW-CRRT study is a prospective, three-arm RCT designed to investigate the hypothesis that higher BFR can prolong the lifespan of extracorporeal circulation in CRRT without anticoagulation.

While anticoagulants are routinely used to extend filter and circuit life in CRRT, certain clinical situations, such as high bleeding risk or altered coagulation function, warrant the exploration of non-pharmacological strategies to improve circuit patency when non-anticoagulation CRRT is indicated. Common non-pharmacologic measures to prolong filter and circuit life include selection of appropriate treatment modality (CVVHD, CVVHDF), mode of dilution, type of filter, management of effluent volume and filtration fraction (<25–30%) [20–22], however these parameters are usually determined by clinical need and institutional experience. Besides, patient's underlying disease, patient comorbidity, intravenous infusion of blood production and nutrients solution are also potential influential factors to filter and circuit longevity in CRRT.

Extracorporeal circulation BFR is a flexible parameter which could be easily adjusted during CRRT. Though recommendations for higher BFR have been proposed to address the short lifespan in clinical practice, there is no conclusive evidence from large-scale RCT studies, and the optimal BFR in CRRT without anticoagulant has yet to be established [13]. From a physiological standpoint, higher BFR is hypothesized to reduce clot formation by shortening the contact time between extracorporeal blood and filters and circuits and minimizing platelet adhesion and activation, as well as coagulation cascade. However, over-highly BFR could exacerbate hemodynamic instability in critically ill patients, posing challenge to filtration efficacy and patient safety [23]. Therefore, determination of the optimal BFR could be related with improved filter and circuit longevity, while its impact on CRRT therapeutic effect and patient adverse events should also be closely monitored.

Literature have reported short filter and circuit lifespans in non-anticoagulation CRRT using low BFR. Zhang *et al.* observed the filter lifespan in anticoagulant-free CRRT at a BFR of 200 mL/min, reporting an overall filter lifespan of 20.5 hours and 16 hours for coagulated filters. The accumulated filter survival probabilities at 12-hour, 24-hour, and 48-hour were 80%, 47%, and 7%, respectively [24]. In another study by Ratanarat. et al. analyzing non-anticoagulation CRRT in critically ill AKI patients with the BFR at 150–200 mL/min, they found that the rate of circuit exchanges due to filter clotting was as high as 71.1% and the median circuit life was only 14.3 hours [9]. According to a single-center, retrospective, observational study where 355 patients and 1332 CRRT treatments were assessed for filter life, higher BFR was related with increased CRRT filter lifespan [25]. However, in an RCT involving 96 AKI patients, a BFR of 150 mL/min was not more likely to cause clotting compared with 250 mL/min [19]. Given the discrepancy of previous studies and the lack of study dedicated to non-anticoagulation CRRT, the FLOW-CRRT trial seeks to advance knowledge by comparing three BFR levels (150, 200, and 250 mL/min), a range chosen based on current clinical practice, on filter and circuit life and other secondary outcomes in patients receiving non-anticoagulation CRRT. The primary aim is to determine whether higher BFRs can mitigate clotting and extend circuit longevity, providing a clinically feasible non-pharmacological strategy where systemic or regional anti-coagulation is contraindicated. It may also help establish evidence-based recommendations on BFR in anticoagulant-free CRRT settings to reduce clotting-related complications and even patient outcomes. Both PP and ITT analyses will be performed. In real-world clinical settings, deviations such as initiating anticoagulation or modifying the BFR during a single CRRT session may occur. Theoretically, whatever anticoagulation modality shifts from anticoagulation-free will aid the prolongation of circuit lifespan. However, such events are expected to occur at a similar rate across the randomized groups.

There are several inherent limitations to this trial that must be acknowledged. First, as a single-center study, the findings may be limited by the lack of regional and ethnic diversity, potentially constraining the generalizability of our findings. A multi-center approach involving population across different countries and regions is essential to validate the findings and applicability. Second, maintaining strict adherence to predefined BFRs may prove challenging in practice due to vascular access dysfunction and hemodynamic instability. Deviations from the intended BFR allocation could impact the accuracy and robustness of the results. Third, this study does not include an economic analysis of CRRT costs associated with extracorporeal circuit clotting, which is an important consideration for healthcare systems. Addressing these gaps will be a priority for subsequent research.

In summary, this study is a prospective RCT exploring the influence of BFR on the extracorporeal circuit life in CRRT without anti-coagulation. By identifying the optimal BFR, the findings might help reduce the frequency of filter and circuit clotting, minimize patient risk, lower healthcare costs, and contribute to the development of evidence-based CRRT protocols for high bleeding risk patients. The findings will also be inspiring for future multi-center studies exploring optimized BFR strategies of CRRT.

## Supporting information

**S1 File. SPIRI checklist.**
(DOC)

**S2 File. Study protocol in Chinese.**
(DOC)

**S3 File. Study protocol in English.**
(DOCX)

## Acknowledgments

None.

## Author contributions

**Conceptualization:** Caihong Liu, Yuliang Zhao.

**Data curation:** Qiongxing Bu, Fang Wang.

**Formal analysis:** Caihong Liu.

**Investigation:** Yongxiu Huang, Jay L Koyner.

**Methodology:** Qiongxing Bu, Wei Wei, Jinglei Ren.

**Writing – original draft:** Caihong Liu.

**Writing – review & editing:** Ling Zhang, Yuliang Zhao.

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
