## [Decision Letter · Decision Letter 0]

25 Jun 2025

PONE-D-25-01201Effects of different blood flow rates on filter and circuit life in non-anticoagulation CRRT: protocol for a three-arm, single-blind randomized controlled trial (the Flow-CRRT Study)PLOS ONE

Dear Dr. Zhao,

Thank you for submitting your manuscript to PLOS ONE. After careful consideration, we feel that it has merit but does not fully meet PLOS ONE’s publication criteria as it currently stands. Therefore, we invite you to submit a revised version of the manuscript that addresses the points raised during the review process.

We look forward to receiving your revised manuscript.

Kind regards,

Wisit Kaewput, MD

Academic Editor

PLOS ONE

Journal Requirements:

5. Please provide a complete Data Availability Statement in the submission form, ensuring you include all necessary access information or a reason for why you are unable to make your data freely accessible. If your research concerns only data provided within your submission, please write "All data are in the manuscript and/or supporting information files" as your Data Availability Statement.

7. Please remove all personal information, ensure that the data shared are in accordance with participant consent, and re-upload a fully anonymized data set.

Additional guidance on preparing raw data for publication can be found in our Data Policy (https://journals.plos.org/plosone/s/data-availability#loc-human-research-participant-data-and-other-sensitive-data) and in the following article: http://www.bmj.com/content/340/bmj.c181.long .

Reviewers' comments:

Reviewer's Responses to Questions

**Comments to the Author**

1. Does the manuscript provide a valid rationale for the proposed study, with clearly identified and justified research questions?

Reviewer #1: Yes

Reviewer #2: Yes

Reviewer #3: Yes

Reviewer #4: Yes

Reviewer #5: Yes

2. Is the protocol technically sound and planned in a manner that will lead to a meaningful outcome and allow testing the stated hypotheses?

Reviewer #1: Yes

Reviewer #2: Yes

Reviewer #3: Yes

Reviewer #4: Yes

Reviewer #5: Yes

3. Is the methodology feasible and described in sufficient detail to allow the work to be replicable?

Reviewer #1: Yes

Reviewer #2: Yes

Reviewer #3: Yes

Reviewer #4: Yes

Reviewer #5: Yes

4. Have the authors described where all data underlying the findings will be made available when the study is complete?

Reviewer #1: Yes

Reviewer #2: Yes

Reviewer #3: Yes

Reviewer #4: No

Reviewer #5: Yes

5. Is the manuscript presented in an intelligible fashion and written in standard English?

Reviewer #1: Yes

Reviewer #2: Yes

Reviewer #3: Yes

Reviewer #4: Yes

Reviewer #5: Yes

6. Review Comments to the Author

You may also provide optional suggestions and comments to authors that they might find helpful in planning their study.

Reviewer #1: As the statisitcal reviewer I will focus on methods and reporting. The trial is generally well design and the approaches are clear. the power calculations are appropriate.

1) mean imputation is not ideal, why wasn't multiple imputation considered instead? and this can be relevant to all modelling choices.

2) univariable and multivariable analyses, not univariate and multivariate

3) make sure you assess the proportional hazards assumption for the Cox models

4) number may be low for controlling for numerous covariates (especially if many are categorical) so the authors need to be ready to prioritise what to control for, if that is the case. Although if there is enough balance in all aspects through the randomisation (as expected since numbers are large for a trial, relatively) a univariable approach would be adequate, at least as a sensitivity analysis.

5) the random effects models are appropriate and well through, but the numbers may be small to safely estimate all components, so again the authors may have to compromise.

Reviewer #2: This is a study proposal. The factors under study are very relevant . The study once completed will give good insight into the problem we routinely encounter

A multicenter study will definitely be better

Reviewer #3: Clarify the impact of protocol deviations, such as BFR adjustments due to clinical needs, on both ITT and PP analyses by providing more information. It would be helpful for clinicians to clarify this.

Despite the thorough statistical approach, certain terminology (such as use of Poisson instead of negative binomial models) may require a succinct explanation for those unfamiliar with count data and overdispersion.

Despite being written in the usual English language, minor modifications could enhance readability (for instance, “staying patent for 24 hours” could be modified to “maintaining it for at least 24 hours”).

Overall, this protocol is technically sound and will provide good data that can inform clinical practice.

Reviewer #4: A randomized controlled trial (RCT) protocol on effects of different blood flow rates on filter life and circuit patency in non-anticoagulation CRRT. this RCT protocol is written in detail and supplementary information file and SPIRIT checklist is complete. I think this protocol satisfies PLOS One publication criteria and is suitable for publication.

I recommend that authors include more information about patients’ safety in the manuscript as they have included in the SI file of trial study protocol.

As per PLOS One Submission Guidelines “the name of the registry and the trial or study registration number must be included in the Abstract”. Moreover, the authors have not described where all data underlying the findings will be made available when the study is complete

Reviewer #5: In their study protocol titled “Effects of different blood flow rates on filter and circuit life in non-anticoagulation CRRT: protocol for a three-arm, single-blind randomized controlled trial (the FlowCRRT Study),” Liu et al. present an important and timely research question. However, I recommend the following revisions to further strengthen the manuscript.

1. Page 8, Introduction Section: The introduction fails to clearly explain the reasoning behind the chosen blood flow rate (BFR) thresholds of 150, 200, and 250 mL/min. As a result, readers may struggle to grasp their clinical significance or the supporting evidence. Including a brief justification based on guidelines or common practice would strengthen the study’s foundation.

2. Page 8, Introduction Section: The authors mention conflicting evidence regarding optimal blood flow rates, with some studies recommending BFR >200 mL/min and others suggesting lower rates may improve circuit patency. However, this important issue is not sufficiently discussed. The introduction would benefit from a clearer synthesis of these differing viewpoints and a more explicit acknowledgment of the ongoing debate in the literature. Highlighting this discrepancy would better justify the need for the current study.

3. Page 9, Study design: The authors should clearly specify how participants will be blinded to treatment assignment in the study design section. While they mention that allocation concealment will be maintained, the exact method of blinding is not described and should be explicitly stated to ensure transparency and reproducibility.

4. Page 9, Study objectives: The authors should avoid vague phrasing and explicitly list the specific secondary outcomes that will be evaluated. Terms like "patient clinical outcomes" and "hemodynamic parameter changes" are too broad, clearly defining which parameters will be assessed is important for the authors to communicate the outcomes effectively.

5. Page 9, Study Participants: The manuscript lacks detail on the screening process for eligible patients. The authors should specify who will be responsible for screening (e.g., nephrologist, ICU physician) and when this will occur during patient care to ensure a clear understanding of the recruitment procedure.

6. Page 9: The authors redundantly mention the blood flow rate (BFR) values in both the Randomization and Intervention sections. This redundancy can be eliminated by presenting the information a single time and making references to it later.

7. PLOS authors have the option to publish the peer review history of their article (what does this mean? ). If published, this will include your full peer review and any attached files.

**Do you want your identity to be public for this peer review?** For information about this choice, including consent withdrawal, please see our Privacy Policy .

Reviewer #1: No

Reviewer #2: **Yes: ** Dr. Archith Boloor

Reviewer #3: **Yes: ** Malak Mohmmed Saed Abdulqadir ( Alagoury )

Reviewer #4: No

Reviewer #5: No

---

## [Author Response · Author response to Decision Letter 1]

21 Jul 2025

Dear Editors and Reviewers:

Thank you for your consideration and for the reviewer’s comments concerning our manuscript entitled “Effects of different blood flow rates on filter and circuit life in non-anticoagulation CRRT: protocol for a three-arm, single-blind randomized controlled trial (the Flow-CRRT Study)” (Manuscript ID: PONE-D-25-01201). The comments were valuable and helpful for revising and improving our manuscript. We have studied the comments carefully with great gratitude and have made corrections and improvements. The detailed point-by-point responses to the reviewer’s comments are as follows. The revised portions are highlighted in RED in the revised manuscript.

I hope that the revised manuscript is suitable for publication in Plos One. I am truly grateful for your help with the manuscript.

Reviewer:

Reviewer #1: As the statistical reviewer I will focus on methods and reporting. The trial is generally well design and the approaches are clear. the power calculations are appropriate.

Comment 1

mean imputation is not ideal, why wasn't multiple imputation considered instead? and this can be relevant to all modelling choices.

Authors’ response: Thanks for the helpful comment. We agree that mean imputation is limited and may lead to biased estimates and underestimation of variance. As you suggested, multiple imputation allows for a more robust and statistically valid approach by incorporating uncertainty arising from missing data. In response, multiple imputation will be used instead of mean imputation to better address this methodological concern. (Page 5, Line 208-210)

Comment 2

univariable and multivariable analyses, not univariate and multivariate.

Authors’ response: Thank you for your kind reminder. We have rectified “univariate” to “univariable” and “multivariate” to “multivariable”. (Page 5, Line 215, 218)

Comment 3

make sure you assess the proportional hazards assumption for the Cox models

Authors’ response: Sincerely appreciate for bringing the proportional hazards assumption to our attention for the Cox models. We will use the Schoenfeld residual test to verify the assumption of proportional hazards in the Cox analysis. In cases of non-proportionality, stratified Cox models or incorporating time-dependent effects will be considered as appropriate. We have added the assessment approach in the revised paper. (Page 5-6, Line 220-222)

Comment 4

number may be low for controlling for numerous covariates (especially if many are categorical) so the authors need to be ready to prioritise what to control for, if that is the case. Although if there is enough balance in all aspects through the randomisation (as expected since numbers are large for a trial, relatively) a univariable approach would be adequate, at least as a sensitivity analysis.

Authors’ response: Thanks for your thoughtful advice. While randomization is expected to ensure baseline balance, the sample size might limit the ability to control for numerous covariates. Therefore, we will prioritize the independently protective or risk factors identified by a retrospective cohort study we previously conducted, including filter type, substitute fluid rate, patients’ coagulation function, hematocrit level, and whether exposure to non-CRRT anti-coagulation agents or concomitant with hyperlipidemia. Then, clinically relevant covariates such as age, gender, and serum creatinine will also be seriously considered if necessary (The retrospective cohort study has not been published yet; we would be pleased to provide the results if needed). Although if it is balanced as expected, we will also conduct a sensitivity analysis with univariable models to examine the robustness of our primary findings. Thanks again for giving such valuable suggestions for improving the rigor of this protocol, which have been added in the revised version. (Page 6, Line 227-233)

Comment 5

the random effects models are appropriate and well through, but the numbers may be small to safely estimate all components, so again the authors may have to compromise. Authors’ response: We sincerely appreciate the reviewer's insightful statistical comments regarding the potential limitations of random-effects models with our sample size. We fully acknowledge that the current data may not provide sufficient power to reliably estimate all variance components in the proposed complex random-effects structure. To address this methodological challenge while maintaining scientific rigor, we will implement a stepwise modeling approach: Initially, we will attempt to fit the pre-specified maximal model incorporating all planned random effects. Should we encounter convergence issues or observe unstable variance estimates (operationally defined as standard errors exceeding the mean estimate), we will systematically simplify the model by first removing random slopes for less critical covariates while preserving random intercepts for key clustering variables like study sites. Throughout this process, we will conduct comprehensive sensitivity analyses to evaluate the impact of model simplifications on both model fit statistics (AIC/BIC) and the stability of fixed-effects estimates, with particular attention to our primary exposure variables. All model adaptations, along with their justifications and validation results, will be thoroughly documented in the supplementary materials to ensure full transparency. (Page 6, Line 241-254)

Reviewer #2: This is a study proposal. The factors under study are very relevant. The study once completed will give good insight into the problem we routinely encounter. A multicenter study will definitely be better

Authors’ response: Thank you for your advice. We will endeavor to conduct a multicenter study once this study is completed and reveals clinically significant insights. We have acknowledged this as a limitation in the Discussion part. (Page 8, Line 327-329)

Reviewer #3:

Comment 1

Clarify the impact of protocol deviations, such as BFR adjustments due to clinical needs, on both ITT and PP analyses by providing more information. It would be helpful for clinicians to clarify this.

Authors’ response: Thank you for your constructive suggestion. When the circuit experiences excessive pressure or triggers pressure alarms at a high BFR, it might be downgraded, which is common in clinical practice. The PP analysis will be performed only for those finished the treatment as planned. Therefore, sessions involving BFR adjustments or changes in anticoagulation modality will be excluded from the PP population to ensure the analysis reflects strict protocol adherence. By contrast, the ITT analysis will include all randomized patients as assigned, regardless of any deviations. We acknowledge that in real-world clinical settings, deviations such as initiating anticoagulation or modifying the BFR during a single CRRT session may occur, typically in response to circuit pressure alarms or clotting risk. Theoretically, whatever anticoagulation modality shifts from anticoagulation-free will aid the prolongation of circuit lifespan. Of note, as the anticoagulant use, the BFR might also be changed (such as when switch to regional citrate anticoagulation). However, such events are expected to occur at a similar rate across the randomized groups. Thank you again for the suggestion, we have enriched discussion part in the revised manuscript. (Page 8, Line 319-323)

Comment 2

Despite the thorough statistical approach, certain terminology (such as use of Poisson instead of negative binomial models) may require a succinct explanation for those unfamiliar with count data and overdispersion.

Authors’ response: Appreciate for your careful review. Poisson regression will be used to model the count data for coagulation frequency, assuming that the mean and variance are equal. If the variance substantially exceeds the mean, a phenomenon known as overdispersion, negative binomial regression may be considered, since it allows for greater flexibility by incorporating a dispersion parameter. (Page 6, Line 255-259)

Comment 3

Despite being written in the usual English language, minor modifications could enhance readability (for instance, “staying patent for 24 hours” could be modified to “maintaining it for at least 24 hours”). Overall, this protocol is technically sound and will provide good data that can inform clinical practice.

Authors’ response: Thanks for your kind suggestions on language, we have modified the corresponding expression to “maintaining it for at least 24 hours” as you suggested. We also thoroughly checked the whole manuscript and made some corrections. (Page 2, Line 54)

Reviewer #4: A randomized controlled trial (RCT) protocol on effects of different blood flow rates on filter life and circuit patency in non-anticoagulation CRRT. this RCT protocol is written in detail and supplementary information file and SPIRIT checklist is complete. I think this protocol satisfies PLOS One publication criteria and is suitable for publication.

Comment 1

I recommend that authors include more information about patients’ safety in the manuscript as they have included in the SI file of trial study protocol. As per PLOS One Submission Guidelines “the name of the registry and the trial or study registration number must be included in the Abstract”.

Authors’ response: Thanks for your advice. Throughout the trial, trained clinical staff will closely monitor adverse events, including hypotension, arrhythmias, bleeding, and other potential complications. if one group exhibits a significantly higher incidence of serious adverse events related to treatment (non-anticoagulation CRRT or BFR setting), such as hypotension, malignant arrhythmia, or death, compared to the other two groups, or if the incidence exceeds predefined safety thresholds, termination of the intervention for that group will be considered immediately. Additionally, an independent Data Safety Monitoring Committee (DSMC) will regularly review trial data to ensure patient safety. Therefore, comprehensive safety monitoring mechanisms have been implemented to effectively protect patient safety throughout the study. We have registered the study protocol at https://www.chictr.org.cn (ChiCTR2400087819) to guarantee it was standardized for patients’ safety and have added it in the Abstract. (Page 1, Line 42; Page 3, Line 104-111)

Comment 2

Moreover, the authors have not described where all data underlying the findings will be made available when the study is complete.

Authors’ response: We thank the reviewer for your careful review. We have now included a Data Availability Statement in the revised manuscript to clarify our plan for data sharing. Specifically, we state that all relevant data from this study will be made available on reasonable request to the corresponding author upon study completion. This addition ensures compliance with PLOS ONE's data availability policy. (Page 9, Line 349-350)

Reviewer #5: In their study protocol titled “Effects of different blood flow rates on filter and circuit life in non-anticoagulation CRRT: protocol for a three-arm, single-blind randomized controlled trial (the FlowCRRT Study),” Liu et al. present an important and timely research question. However, I recommend the following revisions to further strengthen the manuscript.

Comment 1

Page 8, Introduction Section: The introduction fails to clearly explain the reasoning behind the chosen blood flow rate (BFR) thresholds of 150, 200, and 250 mL/min. As a result, readers may struggle to grasp their clinical significance or the supporting evidence. Including a brief justification based on guidelines or common practice would strengthen the study’s foundation.

Authors’ response: Thank you for your constructive suggestions. The Kidney Disease Improving Global Outcomes (KDIGO) consensus guidelines outlined 150-250 mL/min is typically prescribed for CRRT modes such as CVVH and CVVHDF but make no recommendations for practice based on evidence. Recent studies also suggested that very high BFRs (e.g., >250 mL/min) may actually shorten circuit life, possibly due to increased circuit pressure. An RCT comparing BFRs of 150 and 250 mL/min in CRRT found no significant difference in circuit clotting risk. In our center, 200mL/min is the most common BFR setting for anticoagulation-free CRRT. Therefore, based on KDIGO guideline, previous studies, and clinical practice, we established the three BFR levels of 150, 200, and 250 mL/min in this trial. We have provided the supporting evidence on the reason of BFR levels setting in the revised manuscript. (Page 2-3, Line 73-91)

Comment 2

Page 8, Introduction Section: The authors mention conflicting evidence regarding optimal blood flow rates, with some studies recommending BFR >200 mL/min and others suggesting lower rates may improve circuit patency. However, this important issue is not sufficiently discussed. The introduction would benefit from a clearer synthesis of these differing viewpoints and a more explicit acknowledgment of the ongoing debate in the literature. Highlighting this discrepancy would better justify the need for the current study.

Authors’ response: We appreciate the reviewer’s thoughtful and important observation.

Some studies recommend BFR >200 mL/min, however, others pointed out that extremely high BFR (over 250mL/min) might increase the circuit pressure and affect circuit patency. Therefore, “controllable” high BFR might benefit both circuit lifespan and pressure better. Moreover, the KDIGO guidelines suggest that BFRs between 150–250 mL/min are typically prescribed for CRRT modalities such as CVVH and CVVHDF, but offer no evidence-based recommendations for optimal values. Given the relatively low quality, substantial heterogeneity, and controversial findings among previous studies, there is insufficient evidence to draw a solid conclusion on optimal BFR in non-anticoagulation CRRT. Moreover, while clinicians often choose among standard BFR settings (150, 200, or 250 mL/min), no study to date has systematically compared circuit lifespan across these three levels in non-anticoagulated CRRT. Therefore, in this parallel-group, single-blind, three-arm RCT, we aimed to compare the impacts of different BFRs on circuit life and other secondary outcomes in patients receiving non-anticoagulation CRRT. In the revised manuscript, we have highlighted this discrepancy to better justify the need for the current study. (Page 2-3, Line 73-91)

Comment 3

Page 9, Study design: The authors should clearly specify how participants will be blinded to treatment assignment in the study design section. While they mention that allocation concealment will be maintained, the exact method of blinding is not described and should be explicitly stated to ensure transparency and reproducibility.

Authors’ response: Thank you for your valuable comment. To ensure proper blinding in our study, the following measures will be implemented: (1) Allocation Concealment: The CRRT machines and settings will be managed exclusively by the trained CRRT staff, who will not disclose blood flow rate (BFR) assignments to patients or their families. (2) Visual Blinding: The BFR setting on the CRRT machine screen will be obscured or made non-interpretable to patients to prevent inadvertent unblinding. We appreciate your suggestion and have updated the manuscript to explicitly clarify these blinding procedures to enhance transparency and reproducibility. (Page 3, Line 121-125)

Comment 4

Page 9, Study objectives: The authors should avoid vague phrasing and explicitly list the specific secondary outcomes that will be evaluated. Terms like "patient clinical outcomes" and "hemodynamic parameter changes" are too broad, clearly defining which parameters will be assessed is important for the authors to communicate the outcomes effectively.

Authors’ response: We appreciate your thoughtful suggestion for improving the explicitness and rigor of the protocol. We clearly changed “patient clinical outcomes” as patients’ in-hospital mortality, hospital and ICU duration, and invasive mechanical ventilation and dialysis-dependency rate. Blood pressure changes and bleeding occurrence will be recorded during CRRT. We have renewed the information in the S

---

## [Decision Letter · Decision Letter 1]

7 Aug 2025

Effects of different blood flow rates on filter and circuit life in non-anticoagulation CRRT: protocol for a three-arm, single-blind randomized controlled trial (the Flow-CRRT Study)

PONE-D-25-01201R1

Dear Dr. Zhao,

We’re pleased to inform you that your manuscript has been judged scientifically suitable for publication and will be formally accepted for publication once it meets all outstanding technical requirements.

Kind regards,

Wisit Kaewput, MD

Academic Editor

PLOS ONE

Additional Editor Comments (optional):

Accept as is.

Reviewers' comments:

Reviewer's Responses to Questions

**Comments to the Author**

1. Does the manuscript provide a valid rationale for the proposed study, with clearly identified and justified research questions?

Reviewer #1: Yes

2. Is the protocol technically sound and planned in a manner that will lead to a meaningful outcome and allow testing the stated hypotheses?

Reviewer #1: Yes

3. Is the methodology feasible and described in sufficient detail to allow the work to be replicable?

Reviewer #1: Yes

4. Have the authors described where all data underlying the findings will be made available when the study is complete?

Reviewer #1: Yes

5. Is the manuscript presented in an intelligible fashion and written in standard English?

Reviewer #1: Yes

6. Review Comments to the Author

You may also provide optional suggestions and comments to authors that they might find helpful in planning their study.

Reviewer #1: I am satisfied with the authors' responses and resulting changes to the paper. I have no further comments to make.

7. PLOS authors have the option to publish the peer review history of their article (what does this mean? ). If published, this will include your full peer review and any attached files.

**Do you want your identity to be public for this peer review?** For information about this choice, including consent withdrawal, please see our Privacy Policy .

Reviewer #1: No

---

## [Editor Report · Acceptance letter]

PONE-D-25-01201R1

PLOS ONE

Dear Dr. Zhao,

I'm pleased to inform you that your manuscript has been deemed suitable for publication in PLOS ONE. Congratulations! Your manuscript is now being handed over to our production team.

Kind regards,

on behalf of

Dr. Wisit Kaewput

Academic Editor

PLOS ONE